# LARA: A Light and Anti-overfitting Retraining Approach for Unsupervised Anomaly Detection

## ABSTRACT

Most of current anomaly detection models assume that the normal pattern remains the same all the time. However, the normal patterns of web services can change dramatically and frequently over time. The model trained on old-distribution data becomes outdated and ineffective after such changes. Retraining the whole model whenever the pattern is changed is computationally expensive. Further, at the beginning of normal pattern changes, there is not enough observation data from the new distribution. Retraining a large neural network model with limited data is vulnerable to overfitting. Thus, we propose a **L**ight **A**nti-overfitting **R**etraining **A**pproach (LARA) based on deep variational auto-encoders for time series anomaly detection. In LARA we make the following three major contributions: 1) the retraining process is designed as a convex problem and can prevent overfitting as well as converge at a fast rate; 2) a novel ruminate block is introduced, which can leverage the historical data without the need to store them; 3) we mathematically and experimentally prove that when fine-tuning the latent vector and reconstructed data, the linear formations can achieve the least adjusting errors between the ground truths and the fine-tuned ones. Moreover, we have performed many experiments to verify that retraining LARA with even limited amount of data from new distribution can achieve competitive F1 Score in comparison with the state-of-the-art anomaly detection models trained with sufficient data. Besides, we verify its light computational overhead.

## CCS CONCEPTS

• **Computing methodologies** → **Anomaly detection**; • **Information systems** → **Web log analysis**.

## KEYWORDS

Anomaly detection, Time series, Light overhead, Anti-overfitting

**ACM Reference Format:**

Anonymous Author(s). 2018. LARA: A Light and Anti-overfitting Retraining Approach for Unsupervised Anomaly Detection. In *Proceedings of Make sure to enter the correct conference title from your rights confirmation emai (Conference acronym 'XX).* ACM, New York, NY, USA, 11 pages. https://doi.org/XXXXXXX.XXXXXXX

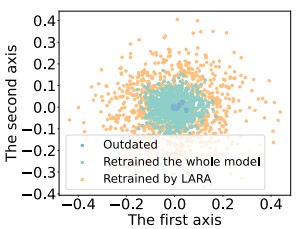

(a) The reconstructed data samples    (b) The latent vectors

**Figure 1: LARA vs. the other two approaches. The figures show the (a) reconstructed data samples and (b) latent vectors outputted by three different approaches: the model trained on historical data only (i.e. outdated model), the model retrained by whole dataset, and the model retrained by LARA.**

## 1 INTRODUCTION

Anomaly detection could dramatically improve the robustness of web services since it can spot various server failures and their root cause efficiently [29, 34]. However, the environment of web services and web servers is highly dynamic [17], where the normal patterns evolve over time, which makes many prominent anomaly detection methods inefficient and inapplicable in this scenario. For example, Recurrent Auto-encoder with Multi-resolution Ensemble Decoding (RAMED) [27] and many other methods achieve remarkable detection accuracy, but they premise that the normal patterns remain the same all the time, which is contradictory to the realistic situation. To keep the high accuracy, these models need to be updated frequently in this scenario, which is inefficient and computational-expensive. Moreover, at the beginning of the distribution shifts, the amount of observed data of new distribution is small, which makes these large networks vulnerable to overfitting. Thus, it calls for a data-efficient and lightweight retraining method.

The existing methods to deal with this problem can be roughly divided into three categories: the signal-processing-based methods, the transfer-learning-based methods, and the few-shot-learning-based methods. Among them, the signal-processing-based methods suffer from heavy inference time overhead and struggle to cope with the high traffic peaks of web services in real time. The transfer-learning-based methods [13] do not consider the chronological order of multiple historical distributions (i.e. the closer historical distribution generally contains more useful knowledge for the newly observed distribution, compared with farther ones). The few-shot-learning-based methods [33] need to store lots of data from historical distributions and also ignore the chronological orders.

To overcome these drawbacks, we propose a Light and Anti-overfitting Retraining Approach (LARA) for deep variational Auto-Encoder-based unsupervised anomaly detection models (VAEs), considering the VAEs are one of the most popular unsupervised

anomaly detection methods. The VAE-based methods learn a latent vector for each input data sample and use the latent vector to regenerate it. The main idea of LARA is to fine-tune the latent vectors with historical and newly observed data without storing the historical data and adapt the reconstructed data samples to the new distribution. This enables LARA to adapt the reconstructed data samples to the center of the new distribution (see Fig.1(a)) and looses the boundary of latent vectors moderately according to the historical and newly observed data (see Fig.1(b)) to enhance the ability of VAEs to deal with unseen distributions. LARA achieves this via three prominent components: the ruminate block, the adjusting functions $M_z$ and $M_x$, and a principle of loss function designing.

The ruminate block[1] leverages the historical data and newly-observed data to guide the fine-tuning of latent vectors, without storing the historical data. The main idea here is that the model trained on historical distributions is an abstraction of their data. Thus, the ruminate block can restore the historical data from the old model and use them with the newly observed data to guide the fine-tuning of the latent vector. There are three advantages of using the ruminate block: 1) it saves storage space as there is no need to store data from historical distributions; 2) it chooses the historical data similar to the new distribution to restore, which would contain more useful knowledge than the others; and 3) with the guidance of the ruminate block, the latent vector generator inherits from the old model and is fine-tuned recurrently, which takes the chronological order of historical distributions into account.

The adjusting functions $M_z$ and $M_x$ are devised to adapt the latent vector and the reconstructed data sample to approximate the latent vector recommended by the ruminate block and the newly observed data respectively. We mathematically prove that linear formations can achieve the least gap between the adjusted ones and the ground truth. It is interesting that the adjusting formations with the least error are amazingly simple and cost light computational and memory overhead. Furthermore, we propose a principle of loss function designing for the adjusting formations, which ensures the convexity of the adjusting process. It is proven that the convexity is only related to the adjusting functions $M_z$ and $M_x$ without bothering the model structure, which makes loss function design much easier. The convexity guarantees the $O(\frac{1}{k})$ converging rate (where $k$ denotes the number of iteration steps) and a global unique optimal point which helps avoid overfitting, since overfitting is caused by the suboptimal-point-trapping.

Accordingly, this work makes the following novel and unique contributions to the field of anomaly detection:

1) We propose a novel retraining approach called LARA, which is designed as a convex problem. This guarantees a quick converging rate and prevents overfitting.
2) We propose a ruminate block to restore historical data from the old model, which enables the model to leverage historical data without storing them and provides guidance to the fine-tuning of latent vectors of VAEs.
3) We mathematically and experimentally prove that the linear adjusting formations of the latent vector and reconstructed data

---

[1]The ruminate block is named after the rumination of cows, which chews the past-fed data and extracts general knowledge.

samples can achieve the least adjusting error. These adjusting formations are simple and require only light time and memory overhead.

In addition, we conduct extensive experiments on four real-world datasets with evolving normal patterns to show that LARA can achieve the best F1 score with limited new samples only, compared with the state-of-the-art (SOTA) methods. Moreover, it is also verified that LARA requires light memory and time overhead for retraining. Furthermore, we substitute $M_z$ and $M_x$ with other nonlinear formations and empirically prove the superiority of our linear formation over the nonlinear ones.

## 2 RELATED WORK

Anomaly detection is to find the outlier of a distribution. We first overview popular anomaly detection approaches for static normal patterns. We then summarize transfer learning, few-shot learning, statistical learning and signal-processing-based methods, which can be used to solve a normal pattern changing problem. There may be a concern that online learning is also a counterpart of LARA. However, the peak traffic of web services is extremely high, and online learning is inefficient and struggles to deal with it in real time.

**Anomaly detection for static normal pattern.** The current popular anomaly detection methods can be divided into classifier-based [7, 9, 14, 23, 24, 26] and reconstructed-based ones [27, 28, 30, 36] roughly. Though these methods achieve high F1 scores for a static normal pattern, their performance decays as the normal pattern changes.

**Transfer learning for anomaly detection.** One solution for the normal pattern changing problem is transfer learning [3, 10, 15, 16, 20, 25]. These methods transfer knowledge from source domains to a target domain, which enables a high accuracy with few data in the target domain. However, these methods do not work well when the distance between source and target domain is large. Moreover, transfer learning mainly transfers knowledge of different observing objects and there is no chronological order of these observations, while the different distributions in this work are observations at different times of the same observing object. The nearer historical distribution is usually more similar to the present one and contains more useful knowledge. But transfer learning ignores this aspect.

**Few-shot learning for anomaly detection.** Few-shot learning determines to extract general knowledge from different tasks and improve the performance on the target task with few data samples. Metaformer [33] is one of the recently prominent few-shot learning based anomaly detection methods, which uses MAML [6] to find an ideal initialization. However, few-shot learning has a similar problem as transfer learning (i.e. it overlooks the chronological orders). Moreover, it needs to store lots of outdated historical-distributed data and costs lots of storage space.

**Statistics-based anomaly detection.** Traditional statistics-based methods [5, 8, 22] do not need training data and have light overhead, which are not bothered by a normal pattern changing problem. However, these methods rely on certain assumptions and are not robust in practice [17].

**Signal-processing-based anomaly detection.** Fourier transform [37] can only capture global information, while wavelet analysis [1] can capture local patterns but is very time-consuming. PCA

**Table 1: The definition of symbols used in this paper.**

| Symbol | Meaning | Symbol | Meaning |
|--------|---------|--------|---------|
| $D_i$ | The $i$th distribution | $V_i$ | The model for $D_i$ |
| $X_i$ | The data samples for $D_i$ | $X_i[j]$ | The $j$th sample in $X_i$ |
| $Z_{i,k}$ | The latent vectors of $X_i$ obtained by $V_k$ | $Z_{i,k}[j]$ | The latent vectors of $X_i[j]$ obtained by $V_k$ |
| $\tilde{X}_{i,k}$ | The reconstructed data of $X_i$ obtained by $V_k$ | $\tilde{X}_{i,k}[j]$ | The reconstructed data of $X_i[j]$ obtained by $V_k$ |
| $\tilde{Z}_i$ | The latent vector of $X_i$ estimated by ruminate block | $\tilde{Z}_i[j]$ | The latent vector of $X_i[j]$ estimated by ruminate block |
| $M_x$ | The adjusting function of reconstructed data | $M_z$ | The reconstructed data of latent vector |
| $\mathcal{P}_x$ | The trainable parameters of $M_x$ | $\mathcal{P}_z$ | The trainable parameters of $M_z$ |

and Kalman Filtering [21] are the most classical signal-processing techniques but are not competitive in detecting anomalies in variational time series. JumpStarter [17] is a recent SOTA method in this category, but it suffers from heavy inferring time overhead and can not deal with a heavy traffic load in real time.

## 3 PROPOSED METHOD

**Preliminary.** VAE-based methods are among the most popular anomaly detection models, such as omniAnomaly [28], ProS [13], Deep Variational Graph Convolutional Recurrent Network (DVGCRN) [4]. These methods are usually divided into two parts: encoder and decoder. An encoder is designed to learn the posterior distribution $p(z|x)$, where $x$ is observed data, and $z$ is a latent vector [12]. A decoder is designed to learn the distribution $p(x|z)$. Please refer to Tab.1 for the definitions of symbols used in this paper.

### 3.1 Overview

The overview of LARA is given in Fig. 2. On the whole, LARA leverages the data from historical distribution and newly-observed data to adjust the latent vector of VAEs and only uses the newly-observed data to adapt the reconstructed data sample. Whenever a retraining process is triggered, LARA firstly uses the ruminate block to restore historical distributed data from the latest model. Then, the ruminate block uses the restored data and newly-observed data to estimate the latent vector for each newly-observed data. Then, LARA employs a latent vector adjusting function $M_z$ to map the latent vector generated by the encoder to the one estimated by the ruminate block. To further adapt the model, LARA also applies an adjusting function $M_x$ to adjust the reconstructed data sample. After that, LARA uses a loss function to fit the trainable parameters in $M_x$ and $M_z$, which guarantees the convexity of the adjusting process.

For each newly-observed data, the ruminate block retrieves $n$ historical data which is similar to the newly-observed data from the latest model. Then, the ruminate block estimates the latent vector for each newly-observed data by Online Bayesian Learning [19], where the restored data and the newly-observed data are regarded as the historical data and the present data in Online Bayesian Learning respectively. The details are given in Sec. 3.2.

The adjusting functions $M_z$ and $M_x$ are mathematically proven to achieve the least adjusting error. Besides, there is an interesting finding that the best formations of $M_z$ and $M_x$ are linear, which are surprisingly simple and require low retraining overhead. The formulation of adjusting error and the mathematical proof is given in Sec. 3.3.

The simple formations of $M_z$ and $M_x$ make it possible to skillfully design a convex loss function for trainable parameters in $M_z$ and $M_x$. The convexity not only prevents the overfitting problem, as there is a unique global optimal point for the convex problem, but also guarantees a fast convergence rate for the retraining process, which contributes to the light time overhead. However, in general, designing such a loss function requires sophisticated techniques and knowledge of convex optimization. Thus, we propose the principles of designing such a convex loss function in Sec. 3.4 to make the designing process easy and convenient. It is found in Sec. 3.4 that the convexity is not related to the model structure but only related to the designing of the loss function, which makes the designing process much easier, without considering the model structure. Any formations that satisfy the principles can guarantee convexity. According to Occam's razor principle [32], LARA chooses one of the simplest as its loss function.

The following subsections illustrate each module, assuming that the retraining process is triggered by newly-observed distribution $D_{i+1}$ and the latest model is $V_i$.

### 3.2 Ruminate block

For each newly-observed sample $X_{i+1}[j]$, the ruminate block firstly obtains the conditional distribution $p(\tilde{X}_{i,i}[j]|Z_{i+1,i}[j])$ from $V_i$. Then, it generates $n$ data samples from the distribution as the historical data. There are two reasons to restore the historical data in this way: 1) the latest model $V_i$ contains the knowledge of historical distribution and its reconstructed data represents the model's understanding of the historical normal patterns; and 2) the historical data is reconstructed from the newly observed data, which selectively reconstructs data similar to the new one among all the historical distributions.

Furthermore, the ruminate block uses the restored historical data $\bar{X}_i$ and newly-observed data $X_{i+1}$ to estimate the latent vector for each newly-observed data. Inspired by the Online Bayesian Learning [19], the expectation and variance of the estimated latent vector, when given a specific $X_{i+1}[j]$, is shown in Eq.1-2, where $\mathbb{E}_z = \mathbb{E}(\tilde{Z}_{i+1}[j]^T \tilde{Z}_{i+1}[j]|X_{i+1}[j])$, $p(z)$ follows the normal distribution, and $\mathbb{E}_{z^T z} = \frac{\mathbb{E}_{z \sim p(z)}[p(\tilde{X}_{i+1,i}[j]|z)p(\bar{X}_i|Z_{i+1})z^T z]}{\mathbb{E}_{z \sim p(z)}[p(\tilde{X}_{i+1,i}[j]|z)p(\bar{X}_i|Z_{i+1})]}$. The proof is given in Appendix. D. The expectations in Eq.1-Eq.2 are computed by the Monte Carlo Sampling [18] and the conditional distributions are given by the decoder of $V_i$.

$$\mathbb{E}(\tilde{Z}_{i+1}[j]|X_{i+1}[j]) = \frac{\mathbb{E}_{z \sim p(z)}[p(\tilde{X}_{i+1}[j]|z)p(\bar{X}_i|Z_{i+1,i})z]}{\mathbb{E}_{z \sim p(z)}[p(\tilde{X}_{i+1}[j]|z)p(\bar{X}_i|Z_{i+1,i})]}, \quad (1)$$

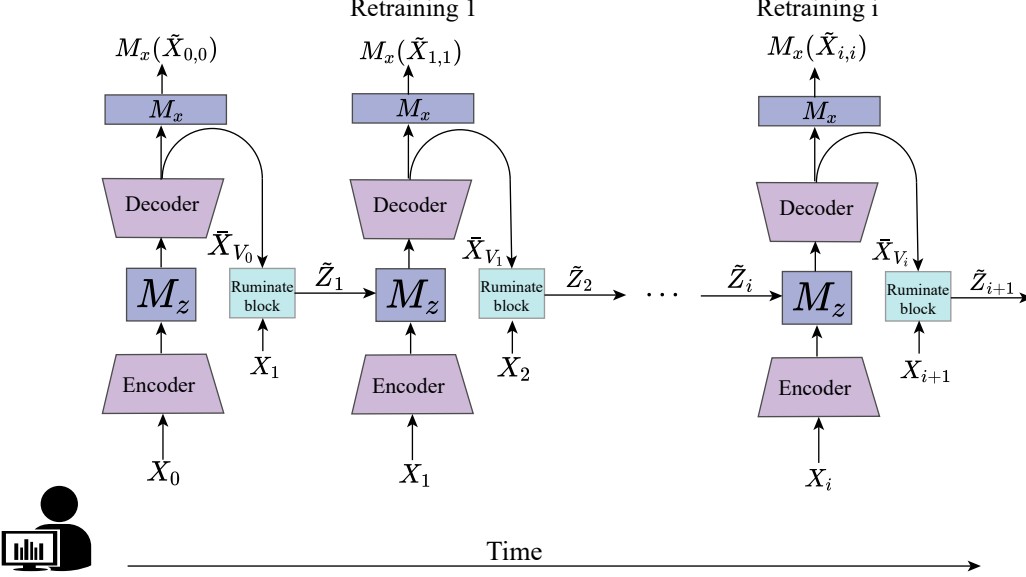

**Figure 2: Overview of LARA. When there is a new distribution shift, LARA retrieves historical data from the latest model and uses them with a few newly observed data to estimate the latent vector for each new sample by the ruminate block. Then, LARA uses two adjusting functions – $M_z$ and $M_x$ – to make two adaptations: adapting the latent vector to the estimated one by the ruminate block, and adapting the reconstructed sample yielded by the latest model to the sample from the new distribution.**

$$\text{Var}(\tilde{Z}_{i+1}[j]|X_{i+1}[j]) = \mathbb{E}_z^T \mathbb{E}_z - \mathbb{E}_{z^T z}. \tag{2}$$

In the following, we provide an intuitive explanation of the Eq.1-Eq.2. We take the $\mathbb{E}(\tilde{Z}_{i+1}[j]|X_{i+1}[j])$ as an example to illustrate it and the variance can be understood in a similar way. When using Monte Carlo Sampling to compute it, the expectation can be transformed into Eq.3, where $\alpha_s = p(\tilde{X}_{i+1,i}[j]|z_s)p(\bar{X}_i|Z_{i+1,i})$ and $z_s$ is the $s$th sampling of $z$ from $p(z)$:

$$\mathbb{E}(\tilde{Z}_{i+1}[j]|X_{i+1}[j]) = \sum_{s=1}^{N} \frac{\alpha_s}{\sum_{k=1}^{N} \alpha_k} z_s. \tag{3}$$

It is now obvious that $\mathbb{E}(\tilde{Z}_{i+1}[j]|X_{i+1}[j])$ is a weighted summation of different $z_s$ throughout the distribution $p(z)$. Next, we look into the value of the weight $\alpha_s$ to figure out under which conditions it assumes higher or lower values. It depends on both the distributions of $p(\tilde{X}_{i+1,i}[j]|z_s)$ and $p(\bar{X}_i|Z_{i+1,i})$. The greater the reconstructed likelihoods of $\tilde{X}_{i+1,i}$ and $\bar{X}_i$ are the greater the weight is. Thus, the estimation of the latent vector is close to the value with a high reconstructed likelihood of $\tilde{X}_{i+1,i}$ and $\bar{X}_i$. As shown in Fig.1(b), this estimation looses the boundary of the latent vector, as it not only considers the reconstructed likelihood of newly observed data, but also considers the one for historical data, which contributes to dealing with the unseen distributions.

## 3.3 Functions $M_x$ and $M_z$

LARA proposes a adjusting function $M_z$ to make $M_z(Z_{i+1,i})$ approximate to $Z_{i+1,i+1}$, which is estimated by $\tilde{Z}_{i+1}$. Moreover, considering each distribution has its specialized features. LARA also uses a adjusting function $M_x$ to make $M_x(\tilde{X}_{i+1,i})$ approximate to $X_{i+1}$.

This subsection solves two questions: 1) what formations of $M_z$ and $M_x$ are the best; and 2) could we ensure accuracy with low retraining overhead? To solve them, the formulation of the adjusting errors of $M_z$ and $M_x$ are firstly given. Then, the formations of $M_z$ and $M_x$ with the lowest adjusting error are explored and proven. We surprisingly find that the best formations are simple and require light overhead.

**Assumption 1.** The $p(\tilde{X}_{i,i}|Z_{i,i})$ and $p(Z_{i,i}|X_i)$ follow Gaussian distribution, which is the same as the assumption in the paper firstly presenting VAE [12].

**Assumption 2.** The $p(Z_{i+1,i}, Z_{i+1,i+1})$ and $p(\tilde{X}_{i+1,i}, \tilde{X}_{i+1,i+1}|Z_{i+1,i})$ follow Gaussian joint distributions, which is reasonable since each marginal distributions follow Gaussian distribution.

**Quantifying adjusting errors of $M_z$ and $M_x$.** We formulate the mapping error $\mathfrak{E}_z$ and $\mathfrak{E}_x$ for $M_z$ and $M_x$ in Eq.(4)-Eq.(5), where $f(X_{i+1})$ and $f(Z_{i+1,i})$ are the probability density functions for $X_{i+1}$ and $Z_{i+1,i}$ respectively:

$$\mathfrak{E}_z = \int \mathbb{E}[(M_z(Z_{i+1,i}) - Z_{i+1,i+1})^2|X_{i+1}]f(X_{i+1})dx, \tag{4}$$

$$\mathfrak{E}_x = \mathbb{E}[(M_x(\tilde{X}_{i+1,i}) - \tilde{X}_{i+1,i+1})^2|Z_{i+1,i}]. \tag{5}$$

The $\mathfrak{E}_z$ actually accumulates the square error for each given $X_{i+1}$ and is a global error, while $\mathfrak{E}_x$ is a local error.

**Theorem 1.** When the Assumptions 1 and 2 hold, the optimal formations of $M_z$ and $M_x$ to minimize $\mathfrak{E}_z$ and $\mathfrak{E}_x$ are as follows:

$$M_z(Z_{i+1,i}) = \mu_{i+1} + \Sigma_{i+1,i}\Sigma_{i,i}^{-1}(Z_{i+1,i} - \mu_i), \tag{6}$$

$$M_x(\tilde{X}_{i+1,i}) = \tilde{\mu}_{i+1} + \tilde{\Sigma}_{i+1,i}\tilde{\Sigma}_{i,i}^{-1}(\tilde{X}_{i+1,i} - \tilde{\mu}_i), \tag{7}$$

where $\mu_{i+1}$ and $\mu_i$ stand for the expectation of $Z_{i+1,i+1}$ and $Z_{i+1,i}$ respectively. $\tilde{\mu}_{i+1}$ and $\tilde{\mu}_i$ stand for the expectation of $\tilde{X}_{i+1,i+1}$ and $\tilde{X}_{i+1,i}$ respectively. $\Sigma_{i+1,i}$ denotes the correlation matrix of $Z_{i+1,i+1}$ and $Z_{i+1,i}$. $\Sigma_{i,i}$ denotes the correlation matrix of $Z_{i+1,i}$ and $Z_{i+1,i}$. $\tilde{\Sigma}_{i+1,i}$ denotes the correlation matrix of $\tilde{X}_{i+1,i+1}$ and $\tilde{X}_{i+1,i}$. $\tilde{\Sigma}_{i,i}$ denotes the correlation matrix of $\tilde{X}_{i+1,i}$ and $\tilde{X}_{i+1,i}$. All of these symbols are trainable parameters of $M_z$ and $M_x$.

*Proof of Theorem 1.* The proof is given in the Appendix B.

As Theorem 1 shows, the linear formations can achieve the least adjusting error and require light retraining overhead.

## 3.4 The principles of retraining loss function

Considering that the formations of $M_z$ and $M_x$ are so simple, it is possible to make the retraining problem convex and its gradient Lipschitz continuous by sophisticatedly designing the loss function. There are two benefits to formulate the retraining problem as a convex and gradient-Lipschitz continuous one: preventing over-fitting (i.e. there is no suboptimal point) and guaranteeing a fast convergence rate $O(\frac{1}{k})$, where $k$ is the iteration steps [2, 31].

Thus, we explore the requirements that the loss function should satisfy to ensure the convexity and gradient-Lipschitz continuousness and find that the convexity of the retraining process is not related to the model structure, but only to the design of the loss function. In the following, the retraining loss function is formally defined in Definition 1 and then Theorem 2 is proposed.

**Definition 1.** As the trainable parameters in retraining stage are only involved in $M_z$ and $M_x$, we define the loss function as $\mathcal{L}(\mathcal{P}_x, \mathcal{P}_z) = \mathcal{L}_x(M_x(\tilde{X}_{i+1,i}; \mathcal{P}_x), X_{i+1}) + \mathcal{L}_z(M_z(Z_{i+1,i}; \mathcal{P}_z), Z_{i+1,i+1})$, where the $\mathcal{L}_x(a, b)$ and $\mathcal{L}_z(a, b)$ are functions evaluating the distance between $a$ and $b$. Besides, $\mathcal{P}_x$ and $\mathcal{P}_z$ stand for trainable parameters in $M_x$ and $M_z$ respectively.

**Theorem 2.** If the two loss functions $\mathcal{L}_x(M_x(\tilde{X}_{i+1,i}; \mathcal{P}_x), X_{i+1})$ and $\mathcal{L}_z(M_z(Z_{i+1,i}; \mathcal{P}_z), Z_{i+1,i+1})$ are convex and gradient-Lipschitz continuous for $M_x(\tilde{X}_{i+1,i}; \mathcal{P}_x)$ and $M_z(Z_{i+1,i}; \mathcal{P}_z)$ respectively, $\mathcal{L}(\mathcal{P}_x, \mathcal{P}_z)$ is convex for $\mathcal{P}_x$ and $\mathcal{P}_z$, and its gradient is Lipschitz continuous. *Proof of Theorem 2.* The proof of Theorem 2 is given in Appendix.

According to Theorem 2, the convexity of the loss function is only concerned with the convexity of $\mathcal{L}_x$ and $\mathcal{L}_z$ for its parameters, without concerning the structures of *Encoder* and *Decoder*. Thus, it is easy to find proper formations of $\mathcal{L}_x$ and $\mathcal{L}_z$ to ensure convexity. According to Occam's razor principle [32], LARA chooses one of the simplest formations satisfied the requirements in Theorem 2 as shown in Eq.(8). $\mathcal{L}_x$ and $\mathcal{L}_z$ can be substituted with any other formation satisfied theorem 2.

$$\mathcal{L}_x = (X_{i+1} - M_x(\tilde{X}_{i+1,i}))^2, \; \mathcal{L}_z = (Z_{i+1,i+1} - M_z(Z_{i+1,i}))^2. \quad (8)$$

## 3.5 Limitation

The ruminate block helps to refresh the general knowledge learned by the old model, but may degrade the accuracy of LARA when the new distribution is very different from the old distribution. We further discuss the impact of distance between old and new distributions on LARA's accuracy in Sec. 4.8.

## 4 EXPERIMENT

Extensive experiments made on four real-world datasets demonstrate the following conclusions:

1) LARA trained by few samples can achieve the highest F1 score compared with the SOTA methods and is competitive against the SOTA models trained with the whole subset (Sec. 4.2).
2) Both $M_z$ and $M_x$ improve the performance of LARA. Besides, the linear formations are better than other nonlinear formations (Sec. 4.3), which is consistent with the mathematical analysis.
3) Both the time and memory overhead of LARA are low (Sec. 4.4).
4) LARA is hyperparameter-insensitive (Sec. 4.5).
5) The experimental results are consistent with the mathematical analysis of the convergence rate (Sec. 4.6).
6) LARA achieves stable performance increase when the amount of retraining data varies from small to large, while other methods' performances suddenly dip down due to overfit and only rebound with a large amount of retraining data (Sec. 4.7).
7) LARA significantly improves the anomaly detection performance for all the distribution-shift distances explored in Sec. 4.8.

## 4.1 Experiment setup

**Baseline methods.** LARA is a generic framework that can be applied to cost-effectively retrain various existing VAEs. In our experiments, LARA is applied to enable two state-of-the-art (SOTA) VAE-based detectors, Donut [34] and OmniAnomaly [28], denoted by LARA-I and LARA-II respectively. We compare LARA with a transfer-learning-based method called ProS [13], a signal-processing-based method called Jumpstarter [17], a classical deep learning method called VAE [12] and the SOTA VAE methods Donut [34] and omniAnomaly [28], Multi-Scale Convolutional Recurrent Encoder-Decoder (MSCRED) [36], AnomalyTransformer [35] and Deep Variational Graph Convolutional Recurrent Network (DVGCRN) [4]. For more details of these baselines, please refer to the Appendix F.

**Datasets.** We use one cloud server monitoring dataset and two web service monitoring datasets. Moreover, to verify the generalization performance, we use one of the most widely recognized anomaly detection benchmark, Soil Moisture Active Passive (SMAP) dataset.

- *Server Machine Dataset (SMD) [28]* is a 5-week-long dataset. It is collected from a large Internet company. This dataset contains 3 groups of entities. SMD has the data from 28 different machines, forming 28 subsets. In this dataset, the data distributions of the training data and retraining data are the most different.
- *Datasets provided by JumpStarter (J-D1 and J-D2) [17]* are collected from a top-tier global content platform. The datasets consist of the monitoring metrics of its 60 different services, forming 60 subsets. In this dataset, the data distributions of the training data and retraining data are the most similar.
- *Soil Moisture Active Passive (SMAP) [11]* is consisted of real spacecraft telemetry data and anomalies from the Soil Moisture Active Passive satellite.

**Evaluation metrics.** We use the widely-used metrics for anomaly detection: precision, recall and F1 score.

**Table 2: Average precision, recall and F1 score results of LARA and baselines. '‡' indicates outdated models that are trained on old distribution data only, while '†' indicates the outdated models are further retrained with small new distribution data.**

| | SMD | | | J-D1 | | | J-D2 | | | SMAP | | |
|---|---|---|---|---|---|---|---|---|---|---|---|---|
| | Prec | Rec | F1 | Prec | Rec | F1 | Prec | Rec | F1 | Prec | Rec | F1 |
| Donut‡ | 0.793 | 0.811 | 0.782 | 0.806 | 0.729 | 0.734 | 0.919 | 0.898 | 0.905 | 0.356 | **1.000** | 0.432 |
| Anomaly Transformer ‡ | 0.304 | 0.654 | 0.415 | 0.331 | 0.852 | 0.471 | 0.842 | 0.986 | 0.907 | 0.297 | **1.000** | 0.456 |
| OmiAnomaly‡ | 0.760 | 0.778 | 0.740 | 0.847 | 0.834 | 0.815 | 0.911 | 0.898 | 0.901 | 0.809 | **1.000** | 0.869 |
| DVGCRN‡ | 0.578 | 0.562 | 0.530 | 0.152 | 0.569 | 0.213 | 0.333 | 0.867 | 0.420 | 0.480 | **1.000** | 0.571 |
| ProS‡ | 0.344 | 0.613 | 0.407 | 0.363 | 0.818 | 0.429 | 0.678 | 0.929 | 0.781 | 0.333 | 0.992 | 0.428 |
| VAE‡ | 0.576 | 0.602 | 0.575 | 0.312 | 0.716 | 0.382 | 0.716 | 0.807 | 0.738 | 0.376 | 0.992 | 0.459 |
| MSCRED‡ | 0.508 | 0.643 | 0.484 | 0.735 | 0.859 | 0.756 | 0.894 | 0.926 | 0.909 | 0.820 | **1.000** | 0.890 |
| LARA-I‡ | 0.613 | 0.885 | 0.697 | 0.815 | 0.650 | 0.682 | 0.828 | 0.969 | 0.891 | 0.400 | **1.000** | 0.493 |
| LARA-II‡ | 0.833 | 0.665 | 0.719 | 0.876 | 0.795 | 0.793 | 0.915 | 0.955 | 0.932 | 0.733 | 0.995 | 0.802 |
| Donut† | 0.742 | 0.795 | 0.764 | 0.950 | 0.650 | 0.727 | 0.906 | 0.913 | 0.904 | 0.502 | **1.000** | 0.578 |
| Anomaly Transformer† | 0.297 | 0.644 | 0.407 | 0.324 | 0.852 | 0.462 | 0.847 | 0.986 | 0.910 | 0.295 | **1.000** | 0.453 |
| OmiAnomaly† | 0.769 | 0.887 | 0.814 | 0.827 | 0.834 | 0.800 | 0.945 | 0.973 | 0.958 | 0.714 | 0.995 | 0.781 |
| DVGCRN† | 0.573 | 0.562 | 0.521 | 0.103 | 0.790 | 0.166 | 0.311 | 0.775 | 0.371 | 0.360 | **1.000** | 0.437 |
| ProS† | 0.504 | 0.533 | 0.415 | 0.375 | 0.732 | 0.373 | 0.758 | 0.803 | 0.769 | 0.574 | 0.992 | 0.620 |
| VAE† | 0.482 | 0.614 | 0.488 | 0.420 | 0.732 | 0.441 | 0.686 | 0.823 | 0.711 | 0.252 | 0.992 | 0.351 |
| MSCRED† | 0.313 | 0.796 | 0.378 | **0.969** | 0.859 | 0.895 | 0.942 | 0.926 | 0.933 | 0.793 | **1.000** | 0.857 |
| LARA-I† | 0.925 | 0.902 | 0.913 | 0.878 | 0.928 | 0.893 | **0.952** | 0.924 | 0.936 | 0.788 | **1.000** | 0.863 |
| LARA-II† | 0.921 | **0.952** | **0.934** | 0.931 | **0.969** | **0.947** | 0.942 | **0.988** | **0.964** | **0.908** | 0.995 | **0.944** |

**Table 3: Compared few-shot LARA with baselines trained with the whole new-distribution dataset.**

| | SMD | | | J-D1 | | | J-D2 | | | SMAP | | |
|---|---|---|---|---|---|---|---|---|---|---|---|---|
| | Prec | Rec | F1 | Prec | Rec | F1 | Prec | Rec | F1 | Prec | Rec | F1 |
| JumpStarter | **0.943** | 0.889 | 0.907 | 0.903 | 0.927 | 0.912 | 0.914 | 0.941 | 0.921 | 0.471 | 0.995 | 0.526 |
| Donut | 0.809 | 0.819 | 0.814 | 0.883 | 0.628 | 0.716 | 0.937 | 0.910 | 0.921 | 0.837 | 0.859 | 0.848 |
| Anomaly Transformer | 0.894 | 0.955 | 0.923 | 0.524 | 0.938 | 0.673 | 0.838 | **1.000** | 0.910 | **0.941** | 0.994 | **0.967** |
| omniAnomaly | 0.765 | 0.893 | 0.818 | 0.914 | 0.834 | 0.855 | 0.918 | 0.982 | 0.947 | 0.736 | 0.995 | 0.800 |
| DVCGRN | 0.482 | 0.611 | 0.454 | 0.214 | 0.599 | 0.256 | 0.412 | 0.867 | 0.444 | 0.410 | **1.000** | 0.478 |
| ProS | 0.495 | 0.623 | 0.418 | 0.210 | 0.760 | 0.306 | 0.566 | 0.886 | 0.688 | 0.287 | 0.992 | 0.395 |
| VAE | 0.541 | 0.728 | 0.590 | 0.353 | 0.550 | 0.392 | 0.686 | 0.823 | 0.711 | 0.416 | 0.992 | 0.473 |
| MSCRED | 0.813 | **0.955** | 0.874 | 0.890 | 0.859 | 0.850 | **0.956** | 0.926 | 0.940 | 0.865 | 0.991 | 0.916 |
| LARA-I† | 0.925 | 0.902 | 0.913 | 0.878 | 0.928 | 0.893 | 0.952 | 0.924 | 0.936 | 0.788 | **1.000** | 0.863 |
| LARA-II† | 0.921 | 0.952 | **0.934** | **0.931** | **0.969** | **0.947** | 0.942 | 0.988 | **0.964** | 0.908 | 0.995 | 0.944 |

## 4.2 Prediction accuracy

All of the datasets used in experiments consist of multiple subsets, which stand for different cloud servers for the web (SMD), different web services (J-D1, J-D2), and different detecting channels (SMAP). Different subsets have different distributions. Thus, the data distribution shift is imitated by fusing the data from different subsets. When verifying the model performances on shifting data distribution, the models are trained on one subset while they are retrained and tested on another one. When using a small amount of retraining data, the models are retrained by 1% of data in a subset. When using enough amount of retraining data, the models are retrained by the whole subset. For each method, we show the performance without retraining, retraining with few samples in Tab.2. Moreover, we also compare the performance of few-shot LARA with the baselines trained with the whole new distribution dataset, which is shown in Tab.3. We obtain precision, recall and F1 score for best F1 score of each subset and compute the average metrics of all subsets. The

"Prec" and "Rec" in Tab.2 stand for precision and recall respectively. For baseline method $\mathcal{A}$, $\mathcal{A}$‡ denotes that $\mathcal{A}$ is trained on the old distribution and tested on the new distribution without retraining. $\mathcal{A}$† denotes that the $\mathcal{A}$ is trained on the old distribution and tested on the new distribution with retraining via a small amount of data from the new distribution and $\mathcal{A}$ denotes that the model is trained on new distribution with enough and much data and tested on new distribution. As JumpStarter is a signal method of sampling and reconstruction, retraining is not applicable to JumpStarter. Besides, transfer between old distribution and new distribution is also not applicable for JumpStarter. Thus, Jumpstarter is not shown in Tab.2. All of the methods retraining with small-amount data use 1% data of each subset from new distribution: 434 time slots of data for SMD dataset, 80 time slots of data for J-D1 dataset, 74 time slots of data for J-D2 dataset, 43 time slots of data for SMAP dataset.

As Tab.2 shows, LARA† achieves the best F1 score on all of the datasets, when compared with baselines retrained with a small

**Table 4: Ablation study results. The best results are in bold, and the second-best results are underlined.**

| | SMD | | | J-D1 | | | J-D2 | | | SMAP | | |
|---|---|---|---|---|---|---|---|---|---|---|---|---|
| | Prec | Rec | F1 | Prec | Rec | F1 | Prec | Rec | F1 | Prec | Rec | F1 |
| remove $M_z$ | 0.855 | **0.973** | 0.892 | 0.913 | 0.861 | 0.879 | 0.842 | 0.975 | 0.901 | 0.751 | **1.000** | 0.833 |
| remove $M_x$ | 0.764 | 0.829 | 0.786 | 0.812 | 0.826 | 0.818 | 0.858 | 0.976 | 0.904 | 0.632 | **1.000** | 0.694 |
| replace with MLP | 0.917 | 0.933 | 0.925 | 0.466 | 0.929 | 0.530 | 0.848 | **0.990** | 0.904 | 0.600 | **1.000** | 0.674 |
| replace with SA | 0.875 | 0.733 | 0.797 | 0.477 | 0.884 | 0.519 | 0.863 | 0.978 | 0.905 | 0.602 | **1.000** | 0.685 |
| LARA | **0.921** | 0.952 | **0.934** | **0.931** | **0.969** | **0.947** | **0.942** | 0.988 | **0.964** | **0.978** | 0.990 | **0.983** |

amount of data. Moreover, LARA[†] also achieves competitive F1 scores when compared with the baselines trained by the whole subset of new distribution, as shown in Tab.3.

Besides, LARA exhibits strong anti-overfitting characteristics, which is also demonstrated in Sec. 4.7. When retrained by small-amount data, LARA[†] dramatically improves the F1 score even with 43 time slots of data, while the F1 score of some baselines reduces dramatically after retraining with small-amount data due to over-fitting.

There is an interesting phenomenon shown in Tab.2. Some methods show better performance when transferred to new distribution without retraining than those trained and tested on new distribution. This verifies that there is some general knowledge among different distributions and it makes sense to utilize a part of an old model.

As LARA-II[†] achieves better performance compared with LARA-I[†], in the following, we look into LARA-II[†] and analyze its performance from different aspects.

## 4.3 Ablation study

We verify the effectiveness of $M_z$ and $M_x$ by separately eliminating them and comparing their performance with LARA. Moreover, to verify the optimality of the linear formation which is proven mathematically in Sec. 3.3, we substitute the linear layer with multi-layer perceptrons (MLP) and self attention (SA) and compare their performance with LARA. The ablation study results are shown in Tab.4. It is clear that 1) removing either $M_z$ or $M_x$ can lead to large F1 score drop on all four datasets, and 2) the linear formation performs better than the non-linear formations, which is consistent with the mathematical analysis in Sec. 3.3.

## 4.4 Time and memory overhead

We use an Intel(R) Xeon(R) CPU E5-2620 @ 2.10GHz CPU and a K80 GPU to test the time overhead. For the neural network methods, we use the profile tool to test the memory overhead for retraining parts of a model. As for JumpStarter, which is not a neural network, we use htop to collect its maximized memory consumption. We show the result in Fig.3(a). From it, we can conclude that LARA achieves the least retraining memory consumption and relatively small time overhead. There are only two methods whose time over-head is lower than LARA: VAE and ProS. But the F1 score of VAE is low. When the distance between source and target domains is large, the F1 score of ProS is low. Moreover, it is also interesting to look into the ratios of retraining time and memory overhead to the training ones, which is shown in Fig.3(b). Since Jumpstarter

is not a neural network and the retraining process is not applica-ble to Jumpstarter, the ratio of Jumpstarter is omitted. As Fig.3(b) shows, LARA achieves the smallest retraining memory ratios and the second smallest retraining time ratios.

## 4.5 Hyperparameter sensitivity

There are two important hyperparameters in LARA: $n$, the number of restored historical data for each data sample from new distribu-tion and $N$, the number of samples when calculating expectation by statistical mean. We test the F1 score of LARA by setting $n$ and $N$ as the Cartesian product of $n$ from 1 to 5 and $N$ from 2 to 10. We show the result in Fig.3(c). Within the search space, the maxi-mum F1 score is only 0.06 higher than the minimum F1 score. With occasional dipping down, the F1 score basically remains at a high level for different hyperparameter combinations. This verifies the hyperparameter insensitivity of LARA.

## 4.6 Convergence rate

To verify the convergence rate of LARA, which is analyzed theo-retically in Sec. 3.4, the loss for each iteration step is divided by $\frac{1}{\sqrt{k}}$, $\frac{1}{k}$, $\frac{1}{k^{1.5}}$, $\frac{1}{k^{1.75}}$ and $\frac{1}{k^2}$ respectively, which is shown in Fig.3(d). The $k$ denotes the iteration steps. As shown in Fig.3(d), when the loss is divided by $\frac{1}{\sqrt{k}}$, $\frac{1}{k}$ and $\frac{1}{k^{1.5}}$, the quotients remain stable as the iteration step grows. Thus, the convergence rate is $O(\frac{1}{k})$, which is consistent with mathematical analysis.

## 4.7 Impact of retraining data amount

We use a subset in SMD to explore the impact of retraining data amount on the performance of LARA†. We firstly use data from one subset to train all of the models. After that, we use different ratios of data from another subset to retrain the models and test them on the testing data in this subset. We show the result in Fig.3(e). When the retraining ratio is 0, there is no retraining. As the figure shows, LARA significantly improves the F1 score even with 1% data from new distribution, while the F1 score of other many methods dramatically dips down due to overfitting. After the first growth of the F1 score of LARA, its F1 score remains stable and high, while the F1 score of many other methods only rapidly grows after using enough data.

## 4.8 Impact of transfer distance between old and new distributions

When training a model on dataset $\mathcal{A}$ and testing it on dataset $\mathcal{B}$, the closer the distributions of $\mathcal{A}$ and $\mathcal{B}$ are, the higher the model's accuracy is. Inspired by this insight, we have defined a

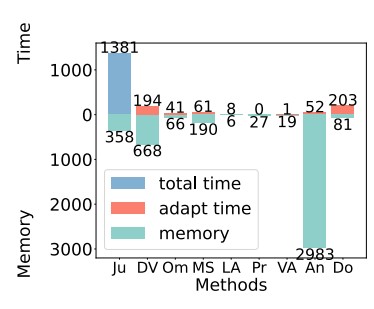

(a) Time (s) and Memory (* 1000B) Overhead for Retraining

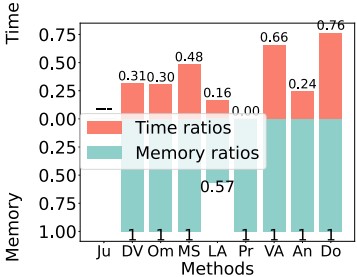

(b) Ratios of Retraining Time (s) and Memory (* 1000B) Overhead to Training Ones

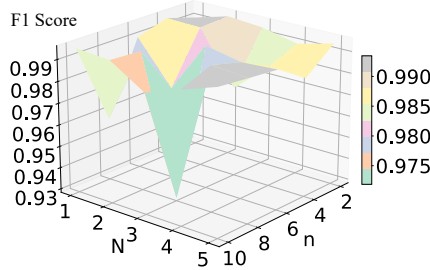

(c) The F1 score for different combination of hyperparameters

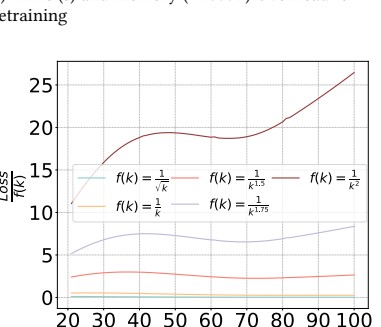

(d) Convergence rate of LARA

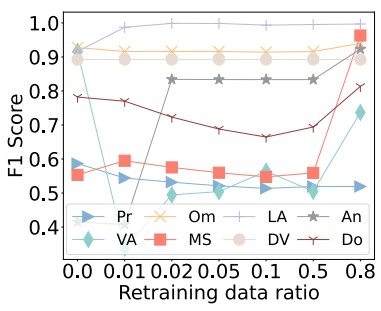

(e) The F1 score for different retraining data amount

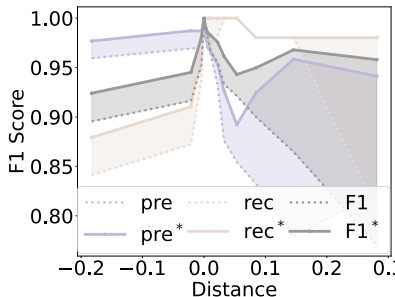

(f) The F1 score for different distribution distance

Figure 3: Due to space constraints, we use the first two letters as the shorthand for each methods. (a) As the memory overhead of JumpStarter, AnomalyTransformer and MSCRED are dramatically larger than the others, to show the memory overhead of other methods clearly, we separately divide the memory overhead of them by 10. (b) The ratios of retraining memory and time overhead to training memory and time overhead. (d) The ratio of loss to iteration steps varies with the number of iteration steps. (e) The x-label is the proportion of retraining data in new distribution data. (f) In the legend, we use pre, rec, F1 to denote the precision, recall and F1 score before retraining and use pre*, rec*, F1* to denote them after retraining.

directed transfer distance to quantify the distance to transfer a specific model $C$ from dataset $\mathcal{A}$ to dataset $\mathcal{B}$ (trained on $\mathcal{A}$ and tested on $\mathcal{B}$) as $Distance_{\mathcal{A}\rightarrow\mathcal{B}}(C)$. This can be quantified by comparing the F1 score degradation when testing on $\mathcal{B}$ after training on $\mathcal{A}$, with testing and training both on $\mathcal{B}$. Let $F1\ score^*$ denote the F1 score for both training and testing on training and testing set of $\mathcal{B}$ and $F1\ score$ denote the F1 score for training on $\mathcal{A}$ and testing on test set of $\mathcal{B}$ without any retraining. Intuitively, the $F1\ score^*$ represents the performance that model $C$ should have achieved on Dataset $\mathcal{B}$ and provides a benchmark for the measurement. We compare the $F1\ score$ with the benchmark and get our transfer distance: $Distance_{\mathcal{A}\rightarrow\mathcal{B}}(C) = \frac{F1\ score^* - F1\ score}{F1\ score}$. Different from traditional distance definition, this distance is asymmetrical ($Distance_{\mathcal{A}\rightarrow\mathcal{B}}(C)$ is different from $Distance_{\mathcal{B}\rightarrow\mathcal{A}}(C)$) and can be negative (for $Distance_{\mathcal{A}\rightarrow\mathcal{B}}(C)$, when the data quality of dataset $\mathcal{A}$ is better than that of dataset $\mathcal{B}$'s training set, and their distributions are very similar, the transfer distance can be negative). Then we show how the transfer distance between old and new distributions impacts the F1 score of LARA in Fig.3(f). Intuitively, when the distance is zero, LARA works best and there is little accuracy difference before and after retraining. Besides, the larger the distance is, the greater the accuracy difference before and after

retraining is. As Fig.3(f) shows, LARA can significantly improve the anomaly detection performance in the range of transfer distance explored in the experiments.

# 5 CONCLUSION

In this paper, we focus on the problem of web-service anomaly detection, when the normal pattern is highly dynamic, the newly observed retraining data is insufficient and the retraining overhead is heavy. To solve these problems, we propose LARA, which is light and anti-overfitting when retraining with little data from the new distribution. There are three distinct characteristics of LARA: it is formulated as a convex problem, a ruminate block, and two light adjusting functions of the latent vector and reconstructed data. The convexity prevents overfitting and guarantees a fast converging rate, which also contributes to a light retraining overhead. The ruminate block makes better use of historical data without storing them. The adjusting functions are mathematically and experimentally proven to achieve the least adjusting errors. Extensive experiments conducted on four real-world datasets demonstrate the anti-overfitting and light properties of LARA. It is shown that LARA retrained with 43 time slots of data is competitive with training the state-of-the-art model with sufficient data.

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

# A  IMPLEMENTATION DETAILS

All of methods retraining with small-amount data on SMD dataset use 434 data samples. All of methods retraining with small-amount data on J-D1 dataset use 80 data samples. All of methods retraining with small-amount data on J-D2 dataset use 74 data samples. All of methods retraining with small-amount data on SMAP dataset use 43 data samples. We mainly use grid search to tune our hyperparameters. The searching range for $n$ is from 1 to 5. The searching range for $N$ is from 1 to 10. The searching range of learning rate is 0.001,0.002,0.005,0.008,0.01. The searching range for batch size is 50,100,400. The searching range of input window length is 40,50,80,100. The searching range of hidden layer is 1,2,3,5.

**Hyperparameters.** The hyperparameters are listed in Tab.5, where $N$ stands for the number of samples when estimating the expectation in Eq.3 and $n$ stands for the number of restored historical data for each newly-observed data sample.

# B  PROOF OF THEOREM 1

*Proof of Theorem 1.* In the following, we take $M_z$ as an example to prove the formation in Eq.(9) is optimal. Then, the formation of $M_x$ can be inferred in the similar way but given $Z_{i+1,i}$ in each step. We

**Table 5: The default hyperparmeter values for LARA.**

| Hyperparameter | Value | Hyperparameter | Value |
|---|---|---|---|
| Batch size | 100 | $n$ | 3 |
| Learning rate | 0.001 | $N$ | 10 |

firstly use the following lemmas 1 and 2 to show that the optimal formation of $M_z(Z_{i+1,i})$ is $\mathbb{E}(Z_{i+1,i+1}|Z_{i+1,i})$. Then, if Assumption 2 holds, we can substitute the $\mathbb{E}(Z_{i+1,i+1}|Z_{i+1,i})$ with the Gaussian conditional expectation and then get the Eq.(9).

$$M_z(Z_{i+1,i}) = \mu_{i+1} + \Sigma_{i+1,i}\Sigma_{i,i}^{-1}(Z_{i+1,i} - \mu_i) \quad (9)$$

**Lemma 1.** The $\mathfrak{E}_z$ can be further transformed into $\mathbb{E}[\mathcal{A}^2] + \mathbb{E}[\mathcal{B}^2]$, where $\mathcal{A}$ and $\mathcal{B}$ are $M_z(Z_{i+1,i}) - \mathbb{E}(Z_{i+1,i+1}|Z_{i+1,i})$ and $\mathbb{E}(Z_{i+1,i+1}|Z_{i+1,i}) - Z_{i+1,i+1}$ respectively.

*Proof of Lemma 1.* According to the definition of expectation, the mapping error can be transformed into $\mathbb{E}_x(\mathbb{E}_{Z_{i+1,i},Z_{i+1,i+1}}((M_z(Z_{i+1,i}) - Z_{i+1,i+1})^2|x))$. For clarity, we use the subscript of $\mathbb{E}$ to denote the variable for this expectation. Furthermore, the two-layer nested expectations can be reduced to the form of a unified expectation $\mathbb{E}_{Z_{i+1,i},Z_{i+1,i+1}}(M_z(Z_{i+1,i}) - Z_{i+1,i+1})^2$. We plus and minus $E(Z_{i+1,i+1}|Z_{i+1,i})$ at the same time and then we get $\mathbb{E}_{Z_{i+1,i},Z_{i+1,i+1}}(\mathcal{A} - \mathcal{B})^2$. We expand the $\mathbb{E}_{Z_{i+1,i},Z_{i+1,i+1}}(\mathcal{A} - \mathcal{B})^2$ and then get $\mathbb{E}[\mathcal{A}^2] - 2\mathbb{E}[(\mathcal{A})(\mathcal{B})] + \mathbb{E}[\mathcal{B}^2]$. We take a further look at the middle term $\mathbb{E}((\mathcal{A})(\mathcal{B}))$ and find it is equal to zero, as shown in the next paragraph. Thus, the Lemma 1 holds.

Now we prove that the $\mathbb{E}[(\mathcal{A})(\mathcal{B})]$ is equal to 0. Since there are two variables in $\mathbb{E}[(\mathcal{A})(\mathcal{B})]$ : the $Z_{i+1,i+1}$ and the $Z_{i+1,i}$, we can transform the $\mathbb{E}[(\mathcal{A})(\mathcal{B})]$ into $\mathbb{E}_{Z_{i+1,i}}[\mathbb{E}_{Z_{i+1,i+1}}[(\mathcal{A})(\mathcal{B})|Z_{i+1,i}]]$. When $Z_{i+1,i}$ is given, the first multiplier in the inner expectation is a constant and can be moved to the outside of the inner expectation. Then we get $\mathbb{E}[(\mathcal{A})(\mathcal{B})] = \mathbb{E}_{Z_{i+1,i}}[\mathcal{A} \cdot \mathbb{E}_{Z_{i+1,i+1}}[\mathcal{B}|Z_{i+1,i}]]$. Recalling $\mathcal{B} = \mathbb{E}(Z_{i+1,i+1}|Z_{i+1,i}) - Z_{i+1,i+1}$, when $Z_{i+1,i}$ is given, $\mathbb{E}(Z_{i+1,i+1}|Z_{i+1,i})$ is a constant and can be moved to the outside of the inner expectation. Then we get $\mathbb{E}[(\mathcal{A})(\mathcal{B})] = \mathbb{E}_{Z_{i+1,i}}[\mathcal{A} \cdot (\mathbb{E}[Z_{i+1,i+1}|Z_{i+1,i}] - \mathbb{E}[Z_{i+1,i+1}|Z_{i+1,i}])]$. Thus, $\mathbb{E}[(\mathcal{A})(\mathcal{B})]$ is equal to 0.

**Lemma 2.** $M_z(Z_{i+1,i}) = \mathbb{E}(Z_{i+1,i+1}|Z_{i+1,i})$ is the optimal solution to minimize $\mathfrak{E}_z$.

*Proof of Lemma 2.* According to Lemma 1, $\mathfrak{E}_z = \mathbb{E}[(M_z(Z_{i+1,i}) - \mathbb{E}(Z_{i+1,i+1}|Z_{i+1,i}))^2] - \mathbb{E}[\mathcal{B}^2]$. Only the first term involves $M_z$. Since the first term is greater than or equal to 0, when $M_z(Z_{i+1,i})$ takes the formation of $\mathbb{E}(Z_{i+1,i+1}|Z_{i+1,i})$, $\mathbb{E}[(M_z(Z_{i+1,i}) - \mathbb{E}(Z_{i+1,i+1}|Z_{i+1,i}))^2]$ reaches its minimum value of 0.

## C PROOF OF THEOREM 2

*Proof sketch of Theorem 2.* We take a further look at the formation of $M_z$ and $M_x$. They can be transformed into affine functions of $\mathcal{P}_z$ and $\mathcal{P}_x$. According to Stephen Boyd [2], when the inner function of a composite function is an affine function and the outer function is a convex function, the composite function is a convex function. Thus, if $\mathcal{L}_x$ and $\mathcal{L}_z$ are convex functions, $\mathcal{L}(\mathcal{P}_x, \mathcal{P}_z)$ is convex. Moreover, as the affine function is gradient Lipschitz continuous, if the $\mathcal{L}_x$ and $\mathcal{L}_z$ are gradient Lipschitz continuous, $\mathcal{L}(\mathcal{P}_x, \mathcal{P}_z)$ is gradient Lipschitz continuous by using the chain rule for deviation.

## D PROOF OF RUMINATE BLOCK

*Proof of Ruminate block.* Since the $\bar{X}_i[j]$ is reconstructed from $X_{i+1}[j]$, it is assumed that they have the approximately same latent vector. It is also assumed that the reconstructed data can approximate the original data. The proof is shown in Eq.10-Eq.11, where $p(z)$ follows the normal distribution which is also assumed by [12].

$$
\begin{aligned}
\mathbb{E}(\tilde{Z}_{i+1}[j]|X_{i+1}[j]) &= \int_z \frac{p(\tilde{X}_{i+1,i}[j]|z)p(\bar{X}_i|Z_{i+1,i})p(z)z}{\int_z p(\tilde{X}_{i+1,i}[j]|z)p(\bar{X}_i|Z_{i+1,i})p(z)dz}dz \\
&= \frac{\mathbb{E}_{z\sim p(z)}[p(\tilde{X}_{i+1}[j]|z)p(\bar{X}_i|Z_{i+1,i})z]}{\mathbb{E}_{z\sim p(z)}[p(\tilde{X}_{i+1}[j]|z)p(\bar{X}_i|Z_{i+1,i})]}
\end{aligned} \quad (10)
$$

$$
\begin{aligned}
\mathbb{E}(\tilde{Z}_{i+1}[j]^T\tilde{Z}_{i+1}[j]|X_{i+1}[j]) &= \int_z \frac{p(\tilde{X}_{i+1,i}[j]|z)p(\bar{X}_i|Z_{i+1,i})p(z)z^Tz}{\int_z p(\tilde{X}_{i+1,i}[j]|z)p(\bar{X}_i|Z_{i+1,i})p(z)dz}dz \\
&= \frac{\mathbb{E}_{z\sim p(z)}[p(\tilde{X}_{i+1,i}[j]|z)p(\bar{X}_i|Z_{i+1,i})z^Tz]}{\mathbb{E}_{z\sim p(z)}[p(\tilde{X}_{i+1,i}[j]|z)p(\bar{X}_i|Z_{i+1,i})]}
\end{aligned} \quad (11)
$$

## E MULTI-RUN EXPERIMENTS FOR DIFFERENT SEEDS

Random seeds introduce significant uncertainty in the training of neural networks. To verify that the results in Tab.2 are not obtained occasionally, we make multi-run experiments for 10 randomly chosen seeds on SMD. The average precisions, recalls and F1 scores are shown in Tab.6. Since Jumpstarter is not a neural network, the multi-seed experiments are not applicable to it. It is proven that the average metrics of multi-run experiments are similar to the results in Tab.2.

## F INTRODUCTION OF THE BASELINES

• *Donut [34]* is one of the prominent time series anomaly detection methods, which improves the F1 score by using modified ELBO, missing data injection and MCMC imputation.

• *OmniAnomaly [28]* is one of widely-recognized anomaly detection methods. It uses a recurrent neural network to learn the normal pattern of multivariate time series unsupervisely.

• *MSCRED [36]* is another widely-recognized anomaly detection method, which uses the ConvLSTM and convolution layer to encode and decode a signature matrix and can detect anomalies with different lasting length.

• *DVGCRN [4]* is a recent method and has reported high F1 score on their datasets. It models channel dependency and stochasticity by an embedding-guided probabilistic generative network. Furthermore, it combines Variational Graph Convolutional Recurrent Network (VGCRN) to model both temporal and spatial dependency and extend the VGCRN to a deep network.

• *ProS [13]* aims at improving the anomaly detection performance on target domains by transferring knowledge from related domains to deal with target one. It is suitable for semi-supervised learning and unsupervised one. For fairness, we use its unsupervised learning as the others are unsupervised.

• *JumpStarter [17]* also aims to shorten the initialization time when distribution changes. It mainly uses a Compressed Sensing technique. Moreover, it introduces a shape-based clustering algorithm and an outlier-resistant sampling algorithm to support multivariate time series anomaly detection.

**Table 6: Average precision, recall and F1 score for 10 seeds on SMD.**

|  | LARA | MSCRED | Omni | ProS | VAE | JumpStarter | DVGCRN | AnomalyTrans | Donut |
|---|---|---|---|---|---|---|---|---|---|
| precision | 0.882 | 0.859 | 0.791 | 0.246 | 0.200 | - | 0.762 | 0.849 | 0.770 |
| recall | 0.971 | 0.967 | 0.904 | 1.000 | 0.754 | - | 0.698 | 0.962 | 0.894 |
| F1 | 0.923 | 0.908 | 0.837 | 0.334 | 0.298 | - | 0.709 | 0.901 | 0.819 |

- *AnomalyTransformer [35]* is one of the latest and strongest anomaly detection methods. It proposes an Anomaly-Attention mechanism and achieves high F1 scores on many datasets.
- *VAE [12]* is one of the classic methods for anomaly detection and is the root of lots of nowadays outstanding methods. It uses stochastic variational inference and a learning algorithm that scales to large datasets.
- *Variants of LARA.* LARA is used to retrain two deep VAE-based methods: Donut [34] and OmniAnomaly [28], which are denoted by LARA-I and LARA-II respectively. To verify the improvement of our retraining approach, we design a controlled experiment. We use LARA-X$^{\ddagger}$ to denote that LARA-X is trained on old distribution and tested on new distribution without retraining, where X can be I or II. We use LARA-X$^{\dagger}$ to denote that LARA-X is trained on old distribution and tested on new distribution with retraining with a small amount of data.

Received 20 February 2007; revised 12 March 2009; accepted 5 June 2009

