# OpenReview forum: "LARA: A Light and Anti-overfitting Retraining Approach for Unsupervised Anomaly Detection"
_ACM.org/TheWebConf/2024/Conference — TheWebConf24_

### Official Review · Reviewer_pBVe · 2023-11-22

**Novelty:** 6
**Technical Quality:** 5

**Review:**

In this paper, the authors propose a distribution-shift-aware retraining model. The main idea of this paper focuses on modeling the training process as an RNN process (like in meta-learning) and proposes a Ruminate block to capture the distribution shift. The Ruminate block is modeled via Monte Carlo sampling. Overall, this is an interesting problem, however, the experiment design is somewhat ad-hoc. The ability of the proposed approach to resist to distribution-shift is unclear. Some experiment settings seem not very clear to me.

Pros:

1. The technique of using meta-learning type process to address distribution-shift is interesting
2. Extensive experiments are performed
3. distribution-shift is an important problem in time series data

Cons:

1. It is unclear whether the proposed task can address distribution-shift (See detailed questions)
2. Some experiment setting is unclear.
3. More backbone model needed to justify the contribution

**Questions:**

1. My major concern is whether the proposed module can capture distribution shifts. It would be nice to show the distribution shift (or some statical of distribution shift) of the tested dataset. From the current description, it is not very clear why these datasets contain distribution shifts.

2. It is unclear how the proposed model detects anomalies. Please add additional information in the supplementary material.

3. Why the base model used is VAE, Donut, and OmniAnomaly? Why are the proposed methods not tested under AnomalyTransformer?

4. It is unclear to me whether the performance mentioned in VAE in Table 3 is trained on new data or not. If it is retrained on new data, I expect the performance can be treated as the golden standard method. However, from the performance, it seems that the performance is not good. Have you compared it with the completely retrained model? If not, it would be nice to show the result.

5. Table 4: Please highlight which LARA model (LARA-I or LARA-II) is used

6. Another issue here is about the ability to capture distribution shift behavior. It would be better to also demonstrate its ability to capture distribution. For example, directly compare with the estimated expectation/variance value and the actual ground truth.

**Reviewer Confidence:**

3: The reviewer is confident but not certain that the evaluation is correct

**Scope:**

4: The work is relevant to the Web and to the track, and is of broad interest to the community

---

### Official Review · Reviewer_dRE9 · 2023-11-23

**Novelty:** 3
**Technical Quality:** 5

**Review:**

This paper targets unsupervised time-series anomaly detection task. Particularly, the authors figure out the phenomenon that the normal patterns of data might change over time, and the trained model may not behave well on the new distribution. Therefore, there exist demands to quickly adapt (finetune/retrain) the trained model to the new distribution.

Motivated by this, authors propose a VAE-based Light Anti-overfitting Retraining Approach (LARA). Specifically, the authors (1) formalize the retraining process as a convex problem, which can prevent overfitting and guarantee fast convergence, and (2) design a ruminate block to leverage historical data (with previous distribution) with low memory requirement.

Pros:
(1) The idea is simple but effective.
(2) The ruminate block is interesting.
(3) Extensive experiments are conducted to validate the effectiveness of LARA.
(4) Good theoretical analysis towards fitting and convergence are provided.

Cons:
See Questions.

**Questions:**

(1) From L86-89, it would be better if the authors could include some practical scenarios to describe the normal/abnormal patterns, which is more friendly to non-expert readers.

(2) (Minor, open question) It is mentioned in L94-95 that "these models need to be updated frequently", and it would be interesting if you can discuss when to update your LARA model (to achieve accuracy and cost balance).

(3) From L137-139, it's a bit strange in the 2nd point, the authors claim there exist similar historical data to the current data to help fine-tuning. However, I have 2 questions: (a) what if the data distribution is suddenly and greatly changed? (I think this case is more common in practical scenarios), and (b) what if the historical and current data distributions are similar, is there any demand for fine-tuning?

(4) From L139-142, if I understand it correctly, "fine-tuned recurrently" means we need to continuously fine-tune the model to keep it up-to-date? It seems impractical and would cost much.

(5) This paper's setting (i.e., training on one data distribution, and testing on another data distribution) is not new, e.g., PUAD [1] has already studied this setting, but PUAD is not included as a baseline.

(6) Regarding other baselines, the latest baselines are Anomaly Transformer (ICLR'22) and DVGCRN (ICML'22). However, by referring to Table 2 and 3, these baselines' performances are very poor, and no explainations are provided. Can you explain the reasons why these latest methods behave much worse than older methods? Besides, more recent baselines are encouraged to be included for comparisons, e.g., [1], [2], [3].

(7) Can you provide some statistics or visualizations about the distribution differences between train and test sets (e.g., of SMD dataset)?

References:
[1] Li, Yuxin, et al. "Prototype-oriented unsupervised anomaly detection for multivariate time series." (2023).
[2] Goswami, Mononito, et al. "Unsupervised model selection for time-series anomaly detection." arXiv preprint arXiv:2210.01078 (2022).
[3] Yang, Yiyuan, et al. "DCdetector: Dual Attention Contrastive Representation Learning for Time Series Anomaly Detection." arXiv preprint arXiv:2306.10347 (2023).

**Reviewer Confidence:**

3: The reviewer is confident but not certain that the evaluation is correct

**Scope:**

3: The work is somewhat relevant to the Web and to the track, and is of narrow interest to a sub-community

---

### Official Review · Reviewer_nQAB · 2023-11-24

**Novelty:** 5
**Technical Quality:** 6

**Review:**

### Summary

This author proposes an approach for unsupervised anomaly detection in time series data, where the normal patterns can change over time. The approach is called LARA, which stands for Light and Anti-overfitting Retraining Approach. LARA is based on deep variational auto-encoders (VAEs) and has the following main features: It uses a ruminate block to leverage historical data without storing them and to guide the fine-tuning of the latent vectors of VAEs. It uses linear adjusting functions to adapt the latent vectors and the reconstructed data samples to the new distribution, proven to achieve the least adjusting errors and low computational overhead. It designs a convex loss function for the retraining process, which ensures fast convergence and prevents overfitting. The paper evaluates LARA on four real-world datasets with evolving normal patterns. It shows that it can achieve competitive F1 scores with few retraining data compared with state-of-the-art anomaly detection models. The paper also verifies LARA's time and memory overhead and its robustness to different hyperparameters and distribution-shift distances.

### Strengths
1. The paper provides mathematical analysis and empirical evidence to support the proposed method. The paper also compares the proposed method with several state-of-the-art baselines and demonstrates its superiority in terms of accuracy, efficiency, and robustness. The paper is well-written and organized, with clear problem formulation, method description, and experimental results.

2. The paper is clear and easy to follow. The paper concisely and intuitively explains the proposed method's main idea and motivation. The paper also provides sufficient details and notations for the technical parts, such as the ruminate block, the adjusting functions, and the loss function design. The paper uses figures and tables to illustrate the method and the results.

3. The paper addresses a challenging and practical anomaly detection problem with evolving normal patterns. The paper proposes a retraining approach that leverages historical and newly observed data to fine-tune the latent vectors and the reconstructed data samples of VAEs. The paper also proves that the linear adjusting functions can achieve the least adjusting error, and the convex loss function can guarantee fast convergence and prevent overfitting. The paper also conducts extensive experiments on four real-world datasets to verify the effectiveness and efficiency of the proposed method.

### Weakness
1. The paper does not provide any code or data to facilitate the reproducibility of the proposed method.
2. The paper should discuss how the performance changes if the same retraining goes through multiple iterations (more on this in the questions sections.).

**Questions:**

1. The paper mentions that the proposed method is applicable to various VAE-based anomaly detection models. Could you elaborate on how the method would need to be adapted for different types of VAEs?

2. The paper does not provide any code or data to facilitate the reproducibility of the proposed method. Could you provide these resources to the research community?

3. Could you discuss the limitations of the proposed method in more detail? Are there any scenarios where the method might not perform well?

4. Have you explored the literature on lifelong and continual learning? Could you explain how continual learning methods compare against the proposed method, and is there any particular reason not to include them in your analysis?  Also, have you encountered catastrophic forgetting in your experiments? If so, how does the proposed method handle it?

5. Have you tried anything similar to the following scenarios? Let's say we have a practical deployment of your model in an industrial plant where the normal data pattern changes every 3 months, and it requires retraining. Now fast forward two years. Your model has to be retrained every 3 months, i.e., 3x4x2=8 times in total. The performance of the proposed model can be slightly lower than the model retrained with sufficient data. How much of a bigger impact might that have on the performance over a long period of time where multiple re-trainings have been performed versus a model that has been retrained from scratch with the whole dataset, including the latest data? This result will show the practicality of the model and if the method can stand the test of time and has the capability to replace current models, which are retrained from scratch every few weeks or months.

**Reviewer Confidence:**

2: The reviewer is willing to defend the evaluation, but it is likely that the reviewer did not understand parts of the paper

**Scope:**

3: The work is somewhat relevant to the Web and to the track, and is of narrow interest to a sub-community

---

### Official Review · Reviewer_Lpht · 2023-11-24

**Novelty:** 4
**Technical Quality:** 4

**Review:**

This paper introduces a lightweight anti-overfitting retraining approach (LARA) based on deep variational auto-encoders to address the challenge of adapting to frequent changes in normal patterns in web services. By featuring a convex retraining process design and the introduction of a novel ruminate block, it effectively tackles issues like overfitting and limited data.

Advantages:
- Solid foundation, clear logic, and outstanding results.

Disadvantages:
- The paper primarily compares itself with transfer learning and few-shot learning, but its standpoint and continuous learning are quite similar.
- The omission of a significant article "CONTINUAL LEARNING FOR ANOMALY DETECTION WITH VARIATIONAL AUTOENCODER", which has utilized VAE for historical sample restoration in the same field, significantly weakens the originality of this paper.

**Questions:**

In comparison with the mentioned article, what is the core innovation of this paper?

**Ethics Review Flag:**

Yes

**Reviewer Confidence:**

2: The reviewer is willing to defend the evaluation, but it is likely that the reviewer did not understand parts of the paper

**Scope:**

3: The work is somewhat relevant to the Web and to the track, and is of narrow interest to a sub-community

---

### Official Review · Reviewer_3CXs · 2023-11-26

**Novelty:** 5
**Technical Quality:** 5

**Review:**

The authors of this paper propose LARA to efficiently prevent overfitting and address the issue of performance degradation in existing unsupervised learning-based time series anomaly detection models, which are not robust to shifts in the trained normality distribution. They demonstrate through various experiments that LARA enables efficient retraining of various VAE-based anomaly detection models.

While designing a time series anomaly detection model robust to distribution shifts is an important research topic, it does not perfectly align with the interests of a Web conference. The paper evaluates the method's performance using cloud server and web service monitoring datasets but lacks an analysis of the actual impact on improving the web ecosystem, aside from the use of web data.

The LARA method proposed in this paper, consisting of the Ruminate block, Adjusting functions, and a retraining loss ensuring convexity, is not entirely novel in this research area. However, it achieves overall performance improvement through appropriate use based on various theoretical analyses. The mathematical and theoretical analysis of each component and design choice in the paper is convincing. However, the notation is overly complex, and the unfriendly formula development makes it very difficult to understand the intended content, necessitating further refinement.

The strengths of this paper include:
- Ensuring technical soundness by presenting theorems and their proofs for all design choices.

- As shown in Figure 3, conducting experiments from a wide range of perspectives to prove the superior performance of LARA and presenting the results in a well-visualized manner.

The weaknesses of this paper are:
- Limited relevance to a Web conference.
- The notation used in the paper is overly complex, making it difficult to understand despite providing various proofs of theorems in the appendix. For example, the continuous use of the notation [j] to represent each sample of each stream data complicates the formula unnecessarily and could be omitted with sufficient explanation. Also, the content in Section 3, based on Figure 2's ‘Retraining i+1’ process, should be aligned with the figure for better understanding.
- All experiments in the Experiment section compare the performance of outdated models, few-shot retrained models, and fully retrained models. Since the problem this paper aims to tackle is creating an anomaly detector robust to distribution shifts, it should also include comparisons with various SOTA methods that directly solve the normal pattern changing problem. The current experiments do not conclusively show that LARA is more efficient at preventing overfitting and retraining compared to other methods.
- The authors mention a limitation of LARA's performance degradation when significantly different distributions are introduced and analyze this in Section 4.8. Although LARA works effectively with datasets where the distance range between distributions is moderate, its performance could drastically decrease if the distance becomes greater. It would have been better if the paper had additionally presented how LARA is more robust or distinctive compared to other methods in solving this endemic problem in all fields addressing distribution shifts.

**Questions:**

In addition to the major points addressed in the weaknesses section of the previous question, here are some minor issues that need attention:

- The second sentence in Section 3.3 appears to be an unfinished sentence. This needs to be revised for clarity and completeness.
- In Table 2, the use of bold and underline should be rechecked. Additionally, it is necessary to clarify what each, bold and underline, signifies in the context of the table.
- In Section 4.8, there seems to be potential for confusion between the newly proposed Distance_(A->B)(C) metric and the x-axis in Figure 3(f). It should be made clear whether these two are related or distinct to avoid any misunderstanding.

**Ethics Review Description:**

no issue

**Reviewer Confidence:**

3: The reviewer is confident but not certain that the evaluation is correct

**Scope:**

2: The connection to the Web is incidental, e.g., use of Web data or API

---

### Decision · Program_Chairs · 2024-01-22

**Decision:**

Accept

**Comment:**

This is the meta-review by the SPC responsible for your paper, and takes into account the opinions expressed by the referees, the subsequent decision thread, and my own opinions about your work.

 - This paper proposes a distribution-shift-aware retraining model called LARA based on deep variational auto-encoders for time series anomaly detection. The authors formulated the retraining process as a convex problem and then introduced a novel ruminate block.

 - Overall, reviewers were impressed by both the mathematical and theoretical analysis, and the extensive experiments from a wide range of perspectives.

 - However, reviewers expressed some concerns, including the limited relevance to the Web Conference and the comparison with continual learning, which the authors addressed during the rebuttal stage and committed to resolving in the final version.